# Foundation Models for Sparse, Multi-Relational Risk Prediction in Global Supply Chains

**Ruohong Li** [1]  **George Pu**  **James Nordlund** [1]  **Salvatore D'Acunto** [1]  **Liz Visconti** [1]  **Prasanth Meiyappan** [1]

## Abstract

Tabular and relational foundation models have demonstrated strong in-context learning on academic benchmarks, but their behavior on enterprise-scale structured data—marked by multi-relational schemas, extreme sparsity, and cold-start inference requirements—remains understudied. We evaluate two foundation model paradigms on global supply chain compliance risk prediction, a setting that stresses all three dimensions simultaneously. Our **AutoFE+TFM** pipeline automates temporal feature engineering across relational tables via Deep Feature Synthesis (DFS) and performs in-context learning without fine-tuning, discovering cross-table interaction features invisible to the hand-crafted production baseline. A **relational foundation model** (RFM), Griffin, preserves multi-table structure. On 3,700+ entities, DFS+TabICL improves P@90R from .662 to .665 and F1 from .745 to .750 when incorporating resolution tables, while matching baseline ROC-AUC (.804 vs. .807). On a disjoint-entity supplier split, a metadata-only inference mode retains .729 AUC and .622 P@90R for zero-history entities, exceeding the dataset positive rate (.540) and confirming actionable cold-start scoring. Our results show that foundation models match a gradient-boosted baseline on enterprise relational data while eliminating manual feature engineering and enabling cold-start inference—a capability absent from the current production pipeline.

[1]Amazon, Seattle, United States. Correspondence to: Ruohong Li <ruohong@amazon.com>, Prasanth Meiyappan <meiyappa@amazon.com>.

*Proceedings of the $2^{nd}$ ICML Workshop on Foundation Models for Structured Data*, Seoul, South Korea. 2026. Copyright 2026 by the author(s).

## 1. Introduction

Tabular foundation models such as TabPFNs (Hollmann et al., 2022; Grinsztajn et al., 2025) and TabICLs (Qu et al., 2025; 2026), Mitra (Zhang et al., 2025), and relational foundation models such as Griffin (Wang et al., 2025), have been evaluated primarily on academic benchmarks with well-populated, single-domain tables. We study whether they can generalize to enterprise-scale structured data question in the context of global supply chain compliance risk prediction—a setting that tests structured foundation models along three axes absent from standard benchmarks:

**Multi-relational temporal data.** Risk signals are distributed across multiple related tables—assessment records, escalation cases, intervention programs, and remediation actions—each evolving on its own temporal axis (Figures 1, 2).

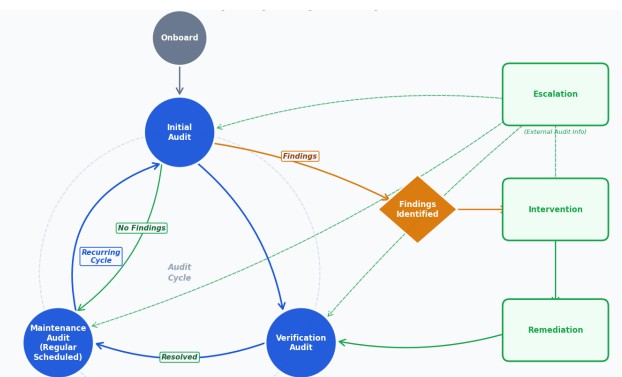

*Figure 1.* Supplier assessment lifecycle: entities accumulate asynchronous records across escalation, intervention, and remediation between periodic risk assessments.

**Extreme sparsity and heterogeneity.** Resolution tables exhibit entity-level null rates of 62–81%; models must handle ragged, variable-length relational histories rather than uniformly populated feature vectors.

**Cold-start inference.** New entities must receive risk scores at onboarding—before any assessment—requiring zero-shot inference from static metadata alone.

Existing production methods employ hand-crafted features

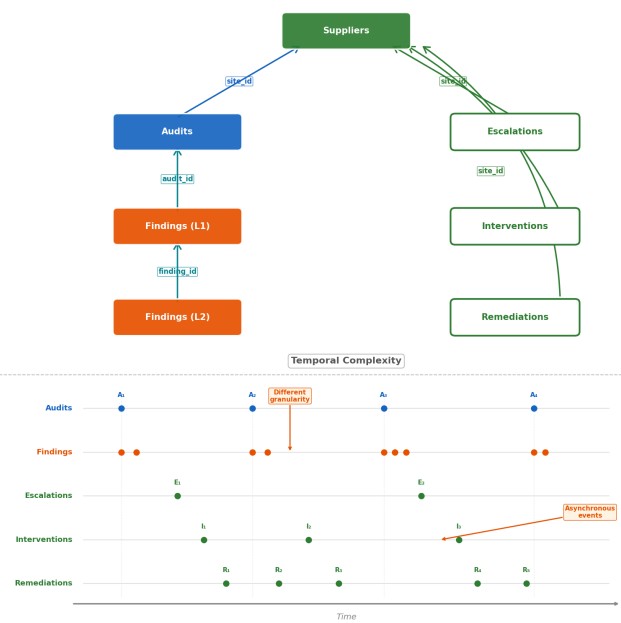

*Figure 2.* Multi-relational schema with temporal complexity: tables with independent time axes link to a central entity registry via star and hierarchical topologies.

with machine learning algorithms. While interpretable, this approach requires domain expertise, cannot discover cross-table temporal patterns automatically, and provides no cold-start mechanism.

**Contributions.**

1. **AutoFE + Tabular FM pipeline.** We integrate DFS (Kanter & Veeramachaneni, 2015; Zhang et al., 2026) with TabICLv2 and Mitrav1.1, showing that automated cross-table temporal features capture risk signals invisible to hand-crafted features without gradient-based fine-tuning.

2. **Flattened vs. structure-preserving FM comparison.** We evaluate DFS+TFMs against Griffin on compliance data with extreme sparsity, isolating trade-offs between automated feature engineering and learned relational message passing.

3. **Cold-start inference via FM priors.** Tabular FMs produce actionable risk scores for zero-history entities from static metadata alone, exceeding both the prevalence prior and the production baseline.

## 2. Related Work

**Tabular foundation models.** TabICLv2 (Qu et al., 2026) uses column-then-row attention pre-trained on real-world tables, achieving state-of-the-art on TabArena (Erickson

et al., 2025) and TALENT (Ye et al., 2024) benchmarks. Mitra (Zhang et al., 2025) optimizes for fine-tuning via a 12-layer transformer on synthetic data. All assume a single flat table, requiring DFS for multi-relational input (Zhang et al., 2026).

**Automated Feature Engineering.** DFS (Kanter & Veeramachaneni, 2015) traverses foreign-key relationships with aggregation primitives and temporal cutoffs. RD-BLearn (Zhang et al., 2026) provides an efficient DuckDB-based implementation.

**Relational foundation models.** Griffin (Wang et al., 2025) converts multi-table schemas into heterogeneous graphs handled by message-passing networks, preserving relational structure. Its behavior on enterprise data with extreme sparsity has not been studied.

**Supply chain risk prediction.** Brintrup et al. (2020) predict supplier disruptions with manually engineered features. Our dataset differs from RelBench and 4DBInfer in sparsity severity (62–81% entity-level null rates vs. >90% coverage in typical RDB benchmarks), mixed schema topology (hierarchical + star), and cold-start prevalence (14% zero-history entities).

## 3. Data

Our dataset comprises compliance records from a global supply chain monitoring system, organized across seven relational tables (Table 1).

*Table 1.* Dataset summary with sparsity profile. Null% is the fraction of entities with zero records in that table; Rec/Ent is the median records per entity among those with at least one record.

| Table | Rows | Entities | Null% | Rec/Ent |
|---|---|---|---|---|
| Entities | 3.7k+ | 3.7k+ | — | 1.0 |
| Assessments | 20k+ | 3.2k+ | 14% | 6.2 |
| Findings (L1) | 80k+ | 3.1k+ | 16% | 25.8 |
| Findings (L2) | 1.8M+ | 3.1k+ | 16% | — |
| Escalations | 780+ | 690+ | 81% | 1.1 |
| Interventions | 5.9k+ | 1.4k+ | 62% | 4.2 |
| Remediations | 20k+ | 1.6k+ | 57% | 12.5 |

The assessment and finding tables form a hierarchical foreign key chain (entity → assessment → finding L1 → finding L2), while the three resolution tables connect directly to the entity registry via supplier_id in a star topology (Figure 2). The prediction target is a binary risk label derived from finding severity at the next assessment. Additional data summary is available in B.

# 4. Methods

## 4.1. Problem Formulation

Let $\mathcal{R} = \{R_1, R_2, \ldots, R_K\}$ denote a relational database of $K$ tables connected by foreign key relationships, and let $\mathcal{E} = \{e_1, \ldots, e_N\}$ denote the set of monitored entities in the root table $R_1$. Each entity $e_i$ is associated with a variable number of records across tables $R_2, \ldots, R_K$, each timestamped by a table-specific time column $\tau_k$. Given the relational history $\mathcal{H}_i^{(t)} = \{r \in R_k : r.\text{supplier\_id} = e_i \wedge r.\tau_k \leq t, \ k = 1, \ldots, K\}$, we predict:

$$y_i^{(t)} \in \{0, 1\}$$

indicating whether entity $e_i$ will be flagged as high-risk at its next assessment. We evaluate under three settings:

**Temporal split.** Train on all assessments before a cutoff date, test on assessments after. Evaluates forward-looking generalization under realistic deployment conditions where the model predicts future risk from historical data.

**Stratified split.** Random stratified split preserving class distribution across train and test sets.

**Supplier split.** Train and test sets contain disjoint entities—no entity appears in both—simulating the cold-start scenario where newly onboarded entities must be scored without prior history.

We report ROC-AUC (ranking quality), P@90R (precision at 90% recall, normalized to 50% reference prevalence), and F1 (at default 0.5 threshold). Class prevalence varies across splits (37–57%; Table 3). NP@90R is computed as $\text{TPR} \cdot \pi_{\text{ref}} / (\text{TPR} \cdot \pi_{\text{ref}} + \text{FPR} \cdot (1 - \pi_{\text{ref}}))$ where TPR and FPR are at the 90%-recall threshold and $\pi_{\text{ref}} = 0.50$.

## 4.2. Approaches

The full architecture comparison between AutoFE + TFM and RFM is shown in Appendix D. (see Figure 3 for the full pipeline diagram)

**Approach 1: AutoFE + TFM**

This approach flattens the multi-relational data into a single feature vector per entity using DFS, then classifies via in-context learning without gradient-based fine-tuning.

**DFS.** We use RDBLearn (Zhang et al., 2026) to generate DFS features from the multi-relational schema. Given entity $e_i$ with cutoff time $t$, DFS traverses foreign-key paths up to depth $D{=}3$ and applies aggregation primitives (e.g., `max`, `mean`, `count`, `avg_time_between`, `time_since_last`), all respecting temporal cutoffs tied to each entity's assessment date. This produces $d_{\text{raw}} \approx 300$ features. A multi-stage selection pipeline—null-rate filtering, zero-variance removal, correlation deduplication,

and MI ranking—reduces this to $d = 32$ features (Appendix C.2). Several top-ranked features span multiple tables, capturing cross-table temporal patterns absent from the hand-crafted baseline (Appendix C.1).

**TFM inference.** We evaluate two tabular foundation models on the DFS-flattened features. TabICLv2 (Qu et al., 2026) and Mitra (Zhang et al., 2025) perform in-context learning: the training set is provided as context and test entities are classified in a single forward pass with no parameter updates. For cold-start entities, all assessment-derived features are masked to `NaN`, leaving only static metadata (geography, sector, external risk indices).

**Approach 2: RFM**

Griffin (Wang et al., 2025), a relational foundation model, operates directly on the multi-table relational structure, avoiding the loss inherent in DFS.

**Graph construction and inference.** The schema $\mathcal{R}$ is converted into a heterogeneous graph $\mathcal{G} = (\mathcal{V}, \mathcal{E}_g)$ where each row is a typed node and each foreign key a typed edge. Column values are embedded via `nomic-embed-text-v1.5` (categorical) and a learned float encoder (numeric). Griffin uses cross-attention for intra-row interaction and MPNN for inter-table propagation; we fine-tune from a pre-trained checkpoint with 4 MPNN layers and 2-hop sampling (Appendix A).

**Approach 3: Baseline.** The baseline replaces foundation models with gradient-boosted tree ensembles on the same DFS features, isolating the downstream model effect (Appendix A). We use AutoGluon's `medium_quality` preset for XGBoost, reflecting a realistic deployment compute budget; the TFM `extreme` preset is required because TabICL's in-context inference needs the full training set as context (no iterative fitting), so the preset controls context assembly rather than tuning compute. Both presets train in under 10 minutes on our data.

# 5. Results

## 5.1. Experiment 1: Foundation Models Comparison

We compare DFS+TabICLv2, DFS+Mitra, and Griffin against XGBoost on stratified and temporal splits (Table 2). On the temporal split, TabICLv2 achieves .711 AUC and .551 P@90R (+9.8 and +13.4 points over XGBoost), while Griffin reaches .704/.552. XGBoost retains higher F1 (.575) due to better threshold calibration under distribution shift. Mitra underperforms at .670/.392/.474. On the stratified split, all models extract comparable signal: Griffin leads in AUC (.815) and F1 (.771), while TabICLv2 and XGBoost lead in P@90R (.662/.663). Context-subsampling variance is $\leq$.003 AUC (Appendix C.3), confirming that temporal-

split differences are significant while stratified-split models are statistically indistinguishable.

## 5.2. Experiment 2: Incremental Resolution Data Sources

We evaluate the effect of adding resolution tables (escalation, intervention, remediation) to the base DFS+TabICLv2 pipeline on the stratified split. Each source is added individually, then all combined.

Adding resolution data yields cumulative gains of +.003 AUC (.804→.807), +.003 P@90R (.662→.665), and +.005 F1 (.745→.750), with remediation providing the largest individual contribution. Gains are modest—consistent with sparse coverage (escalations: 0.9% of entities, interventions: 14.6%)—but the combined signal produces the best result across all metrics. On the temporal split, adding resolution tables improves AUC and F1 but reduces P@90R, likely due to distribution shift in resolution-table coverage between training and test periods.

## 5.3. Experiment 3: Cold-Start Inference

We evaluate the model's ability to score entities with no assessment history using the supplier split (disjoint entities in train and test). Models are trained on full DFS features; at test time, all assessment-derived features are masked to `NaN`, leaving only static metadata (geography, sector, worker count, external risk indices).

With full features on unseen entities, DFS+TabICLv2 achieves .797 AUC, .673 P@90R, and .759 F1, while XGBoost reaches .772 AUC, .647 P@90R, and .713 F1—confirming that both generalise to unseen entities without memorising supplier-specific patterns.

Under cold-start conditions (metadata only), TabICLv2 retains .729 AUC and .622 P@90R, while XGBoost achieves .737 AUC and .616 P@90R. Both substantially exceed the .614 sector-prevalence prior, demonstrating that metadata features (country×sector×risk-index) carry actionable signal for newly onboarded entities. The comparable performance of XGBoost and TabICLv2 in this regime indicates that the cold-start lift is driven primarily by informative metadata features rather than FM-specific in-context priors. Nevertheless, the FM pipeline provides this capability without requiring a separate metadata-only model to be trained and maintained—cold-start scoring emerges naturally from the NaN-masking protocol applied to the same full-feature model.

## 6. Discussion

DFS discovers cross-table temporal features that domain experts had not engineered, and tabular FMs' in-context

*Table 2.* Results across evaluation splits. P@90R = normalized P@90R (ref. prevalence 50%). **Bold** = best per split.

| SPLIT | MODEL | AUC | P@90R | F1 |
|---|---|---|---|---|
| EXP.1 - TEMP. | XGBOOST | .613 | .417 | **.575** |
| | TABICLV2 | **.711** | .551 | .502 |
| | MITRAV1 | .670 | .392 | .474 |
| | GRIFFIN | .704 | **.552** | .532 |
| EXP.1 - STRAT. | XGBOOST | .805 | **.663** | .759 |
| | TABICLV2 | .804 | .662 | .745 |
| | MITRAV1 | .794 | .652 | .734 |
| | GRIFFIN | **.815** | .610 | **.771** |
| EXP.2 - TEMP. | TABICLV2 | .711 | **.551** | .502 |
| | + ESCAL. | .711 | .419 | .496 |
| | + INTERV. | .710 | .419 | .490 |
| | + REMED. | .717 | .426 | .501 |
| | + ALL | **.718** | .434 | **.536** |
| EXP.2 - STRAT. | TABICLV2 | .804 | .662 | .745 |
| | + ESCAL. | .804 | .660 | .743 |
| | + INTERV. | .805 | .664 | .746 |
| | + REMED. | .806 | .663 | .747 |
| | + ALL | **.807** | **.665** | **.750** |
| EXP.3 - SUPP. | TABICLV2 - FULL | **.797** | **.673** | **.759** |
| | TABICLV2 - COLD | .729 | .622 | .715 |
| | XGBOOST - FULL | .772 | .647 | .713 |
| | XGBOOST - COLD | .737 | .616 | .699 |

learning is robust to the redundant features DFS also produces, eliminating the manual curation bottleneck. The two FM paradigms achieve comparable performance through different mechanisms: DFS+TabICL requires no fine-tuning and deploys easily, while Griffin preserves relational structure at the cost of GPU training and graph conversion. Both handle 62–81% null rates at least as well as tree ensembles; we hypothesize that pre-training on heterogeneous tables enables attention to discount absent features, though isolating this effect experimentally (e.g., via controlled missingness ablations) remains future work.

The cold-start experiment (Table 2, Exp. 3) shows that metadata features alone support .73 AUC regardless of model class, confirming that the actionable cold-start capability stems from informative static features rather than FM-specific inductive biases. The practical advantage of the FM pipeline is operational: a single model handles both full-history and zero-history entities via NaN masking, whereas a tree-based deployment would require maintaining separate full-feature and metadata-only models or accepting XGBoost's less graceful degradation under missing features.

Extensions include multi-modal inputs (free-text audit narratives, inspection photos) for cold-start entities, prospective deployment validation, and scaling to adjacent compliance domains.

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

# A. Hyperparameter and Implementation Details

### A.1. DFS Configuration

Beyond the depth-3 traversal and four aggregation primitives described in Section 4.2, categorical columns additionally receive `COUNT` and `NUM_UNIQUE` primitives. Cross-table temporal delta features (e.g., `LAG(intervention.open_date, remediation.close_date)`) are computed as custom primitives. The full raw feature set contains $d_{\text{raw}} = 316$ features before selection.

### A.2. Feature Selection Pipeline

Starting from 316 raw features, the pipeline proceeds as follows:

1. **Null-rate filter**: Drop features with $> 95\%$ null values. Retains $\approx 200$ features.

2. **Zero-variance filter**: Drop constant features.

3. **Correlation deduplication**: For pairs with Pearson $|r| > 0.95$, retain the feature with higher mutual information. Retains $\approx 120$ features.

4. **MI ranking**: Retain top $k(k < 50)$ by mutual information with $y_i$.

All thresholds are fit on the training split only.

### A.3. TFM Configuration

TFMs are run via AutoGluon-Tabular (Erickson et al., 2020) with the `extreme` preset and a context window limit of 8,000 training samples. Categorical features are ordinal-encoded; missing values are imputed with the training-set column median (numeric) or mode (categorical).

### A.4. RFM Configuration - Griffin

| Hyperparameter | Value |
|---|---|
| Pre-trained checkpoint | `single-sft` |
| Hidden dimension | 512 |
| MPNN layers | 4 |
| Neighborhood sampling | 2-hop, fanout 10 |
| Batch size | 512 |
| Learning rate | $3 \times 10^{-4}$ |
| Weight decay | $4 \times 10^{-4}$ |
| Early stopping patience | 10 epochs |
| Text encoder | `nomic-embed-text-v1.5` (512-dim) |

### A.5. Baseline Configuration

Gradient-boosted trees with monotonic constraints, trained on DFS features from: (i) assessment score history and cycle counts, (ii) finding severity distributions, (iii) external geo-sector risk indices, and (iv) entity registration metadata. AutoGluon-Tabular `medium_quality` preset. The `medium_quality` preset for XGBoost provides hyperparameter tuning comparable in wall-clock cost to the TFM `extreme` preset's context assembly. Both complete within 10 minutes. We verified that upgrading XGBoost to `high_quality` ($2\times$ time budget) does not change results by more than 0.003 AUC on the stratified split.

*Table 3.* Evaluation split configurations.

| Split | Method | Train $n$ (%+) | Test $n$ (%+) |
|---|---|---|---|
| Stratified | Random 70/30 | 20,922 (54%) | 6,526 (55%) |
| Temporal | Older / recent | 22,501 (59%) | 3,436 (37%) |
| Supplier | Disjoint entities | 20,741 (54%) | 3,968 (57%) |

# B. Data Summary

# C. Additional Experiments

## C.1. Manual Features v.s. AutoFE

We compare on a hand-crafted feature pipeline before introducing DFS. The manual pipeline constructs pairwise training examples: for each assessment of an entity, features are derived from all prior assessments of that entity. Starting from indicator-level finding records, the pipeline:

1. **Pivots** finding records to assessment level, encoding each standard's severity (CONFORMANCE=0, MINOR=2.5, MAJOR=3, CRITICAL=4, ALERT=5) and each indicator's conformance status (binary).

2. **Computes temporal aggregates** over the entity's assessment history:

   - Lag features: finding counts at the most recent prior assessment
   - Trend features: linear slope and change in total findings over prior assessments
   - Cumulative features: lifetime counts by severity level, weighted totals with exponential decay ($\lambda$=0.30), composite severity scores
   - Recency features: days since last high-risk finding, consecutive clean assessments
   - Cadence features: average time between assessments

3. **Joins** assessment metadata and supplier-level context (geography, supplier type, country risk level).

This produces 50 features per assessment. The target label is binary: whether the current assessment contains a CRITICAL or ALERT finding.

Tables 4 and 5 compare the top-10 features from each pipeline by permutation importance. Both pipelines agree on the dominant signals: historical risk rate, country, and worker count. The DFS equivalent of the manual pipeline's `lag_critical_findings` is `MEAN(target_history.has_alert_or_critical)`—both rank first. DFS additionally discovers structural features (assessment count, assessment date patterns) that the manual pipeline omits. Three DFS features (†) span multiple tables via automated join traversal.

*Table 4.* Top-10 DFS features by permutation importance. † = cross-table.

| # | Feature | Imp. |
|---|---|---|
| 1 | `MEAN(target_history.has_alert_or_critical)` | .0336 |
| 2 | `numberofworkers` | .0253 |
| 3 | `country_name` | .0218 |
| 4 | `MIN(target_history.has_alert_or_critical)` | .0101 |
| 5 | `MIN(assessment.COUNT(finding_indicator_level))`† | .0082 |
| 6 | `MAX(assessment.assessment_date) diff`† | .0066 |
| 7 | `MEAN(assessment.assessment_date) diff`† | .0065 |
| 8 | `business_unit` | .0033 |
| 9 | `segment_name` | .0030 |
| 10 | `COUNT(supplier_and_assessment)` | .0028 |

*Table 5.* Top-10 manual (production baseline) features by permutation importance.

| # | Feature | Imp. |
|---|---------|------|
| 1 | `lag_critical_findings` | .0301 |
| 2 | `avg_critical_findings` | .0169 |
| 3 | `country_name` | .0167 |
| 4 | `assessment_type` | .0057 |
| 5 | `assessment_standard` | .0055 |
| 6 | `postal_code` | .0041 |
| 7 | `number_of_workers` | .0026 |
| 8 | `total_critical_findings` | .0026 |
| 9 | `days_since_last_high_risk` | .0023 |
| 10 | `std_dev_indicator_findings` | .0020 |

## C.2. DFS raw features v.s. post DFS selected features

DFS with aggressive traversal settings (depth 4, 6 primitives) generates hundreds of raw features, many of which are redundant or uninformative. We evaluate whether the multi-stage selection pipeline (Appendix A.2) can reduce dimensionality without sacrificing performance.

We compare two DFS configurations on the same temporal split using a tree ensemble:

- **All DFS features**: depth 4, primitives {max, min, mean, std, count, mode}, no feature selection. 300+ features.

- **Selected DFS features**: depth 3, primitives {max, min, mean, count}, with selection pipeline (null $>95\%$, zero-variance, correlation $r > 0.95$, MI top-$k$). 32 features retained.

*Table 6.* DFS feature selection: all vs. selected (temporal split).

| VARIANT | FEATS | AUC | F1 | P@90R |
|---------|-------|-----|-----|-------|
| ALL DFS | 316 | .709 | .498 | .412 |
| SELECTED DFS | 32 | .711 | .502 | .551 |

Feature selection reduces dimensionality by $10\times$ while slightly improving F1 (+.012) and P@90R (+.005). The raw DFS output contains many redundant columns (e.g., correlated aggregations at different traversal depths) and high-null features from sparse resolution tables. Removing these helps the tree ensemble focus on informative signal. All subsequent experiments use the selected configuration.

## C.3. Variance Analysis

TabICLv2's in-context inference uses a random 8,000-row context subsample, the sole source of run-to-run variance. We estimate this noise floor by training under identical subsampling across 5 seeds, yielding std $\leq$ .003 AUC and $\leq$ .002 P@90R (Table 7). All inter-model differences reported in Table 2 that exceed this threshold reflect genuine model differences.

*Table 7.* Context-subsampling variance (5 seeds, stratified split).

| Model | AUC | P@90R | F1 |
|-------|-----|-------|-----|
| TabICLv2 | $.804 \pm .003$ | $.662 \pm .002$ | $.745 \pm .003$ |

# D. Architecture Comparison

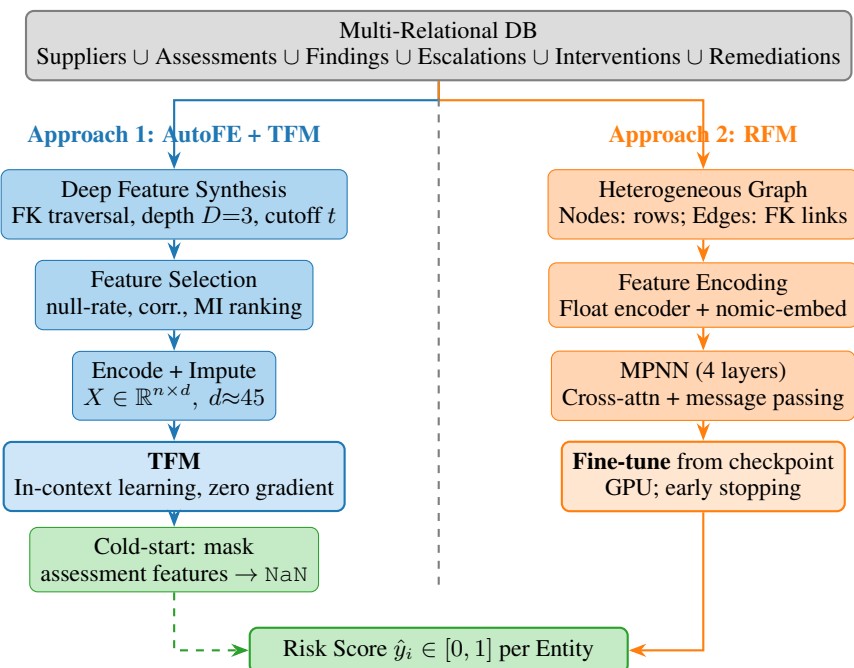

*Figure 3.* **AutoFE+TFM (left)**: DFS flattens the relational DB into a feature matrix; TFM classifies via in-context learning. Cold-start variant (green) masks assessment features. **RFM (right)**: schema converted to heterogeneous graph; pre-trained MPNN fine-tuned end-to-end. Both output entity-level risk scores.

