# OpenReview forum: "Foundation Models for Sparse, Multi-Relational Risk Prediction in Global Supply Chains"
_ICML.cc/2026/Workshop/FMSD — FMSD @ ICML 2026 Poster_

### Official Review · Reviewer_Gv9Q · 2026-05-18
**Relevant application, but empirical claims are stronger than the evidence supports**

**Rating:** 6
**Confidence:** 4

**Review:**

### Summary

This paper studies foundation-model approaches for sparse, multi-relational supply-chain compliance risk prediction. The authors combine Deep Feature Synthesis with tabular foundation models such as TabICLv2 and Mitra, and compare this flattened AutoFE+TFM pipeline against Griffin, a relational foundation model, and an XGBoost baseline. The paper evaluates performance under stratified, temporal, and supplier-disjoint/cold-start settings. The application is well aligned with the workshop’s interest in real-world structured-data foundation models, especially because the dataset involves relational schemas, temporal histories, high sparsity, and cold-start inference requirements.

### Strengths

The paper addresses a highly relevant and practical use case. Supply-chain risk prediction is a strong example of an enterprise structured-data problem where relational structure, sparsity, and temporal leakage risks matter.

The comparison between flattened DFS-based tabular foundation models and a structure-preserving relational foundation model is interesting and appropriate for the workshop theme.

The paper includes several useful evaluation settings, especially the temporal split and supplier-disjoint split, which are more realistic than a purely random stratified split.

The cold-start setting is particularly relevant, since newly onboarded suppliers may need to be scored before historical assessment data exists.

### Areas for improvement

The main concern is that some claims are stronger than the experimental evidence supports. In Experiment 1, TabICLv2 and Griffin do outperform XGBoost on AUC and P@90R under the temporal split, but XGBoost achieves the best temporal F1 and remains highly competitive on the stratified split. This makes it difficult to conclude broadly that foundation models outperform tree ensembles. A more careful framing would distinguish ranking performance from thresholded classification performance.

The claim that foundation models handle high null rates better than tree ensembles is not fully isolated experimentally. The paper reports high sparsity in the relational tables, but does not provide an ablation by sparsity level or missingness pattern. Since the selected DFS features with a tree ensemble are already competitive, the paper should either provide stronger evidence for this claim or soften it.

The abstract highlights gains from adding resolution tables, but these appear to be within-model ablations of DFS+TabICL rather than improvements over an external or production baseline. The reported improvements are also very small. More importantly, in the temporal split, adding all resolution tables appears to reduce P@90R substantially, which is not discussed. This weakens the use of Experiment 2 as headline evidence.

The cold-start result is interesting, but the interpretation is under-supported. The metadata-only setting may still contain strong predictors such as geography, sector, external risk indices, and supplier attributes. A metadata-only XGBoost or logistic regression baseline would be needed before attributing the strong cold-start performance to foundation-model priors or in-context learning.

The metric choice needs clearer justification. P@90R seems especially appropriate for supply-chain risk screening, where high recall is important but false positives affect assessment workload. ROC-AUC and F1 are also useful, but they answer different operational questions. The paper reports all three without sufficiently explaining their relative importance. The thresholding/calibration procedure for F1 should also be specified, since model rankings change depending on the metric.

### Detailed comments

- Please define P@90R/NP@90R when it is first introduced. It is unclear whether this is raw precision at 90% recall or precision normalised relative to class prevalence, and the notation is inconsistent between the main text and Table 2.
- Please explain how thresholds are selected for F1, especially under temporal shift.
- Please include a metadata-only classical baseline for the cold-start experiment.
- Please clarify the baseline in the abstract. The phrase “matching baseline ROC-AUC” could be read as referring to an external baseline, whereas the comparison appears to be DFS+TabICL with fewer tables.
- Please discuss why adding resolution tables improves stratified metrics slightly but appears to hurt temporal P@90R substantially. In Table 2, the temporal P@90R drops from .551 for TabICLv2 to .434 when all resolution tables are added, yet .434 appears to be bolded as if it were the best result in that block. This seems to be either a reporting/formatting error.
- There are minor presentation issues, including typos in Table 1 such as “Row/Enttity”.
- Some reported numbers appear inconsistent between the Experiment 3 text and Table 2; these should be checked.
- Please clarify the fairness of the AutoGluon preset choices. The TFM models are run with the extreme preset, while the XGBoost baseline uses the medium quality preset. Since XGBoost is already highly competitive in several settings, it would be useful to know whether the comparison holds under comparable tuning/compute budgets.

### Justification of score

I recommend acceptance at the margin. The paper is clearly relevant to the workshop and studies an interesting real-world structured-data problem. The experimental setup is promising, and the cold-start setting will likely be of interest to the workshop audience. However, the current version overstates several conclusions, especially around foundation-model superiority, sparsity handling, and the interpretation of the cold-start results. I am also concerned by inconsistencies in the reported results, particularly the mismatch between the Experiment 3 text and Table 2, as well as the apparent drop in temporal P@90R when adding resolution tables, which is neither discussed nor correctly highlighted in the table. Since some of the paper’s main findings rely on these reported numbers, these issues should be resolved before the claims can be fully trusted.

---

### Official Review · Reviewer_N9ae · 2026-05-22
**A well-motivated, in-scope case study showing tabular FMs match XGBoost-on-DFS and enable cold-start on sparse multi-relational supply-chain data, but headline claims outrun the evidence (no variance, "match production" never measured, AUC-vs-prevalence mix-up, several text/table inconsistencies).**

**Rating:** 6
**Confidence:** 3

**Review:**

## Summary of contributions
The paper asks whether tabular and relational foundation models can transfer to enterprise-scale structured data that is multi-relational and sparse. As a testbed, it uses risk prediction for compliance in global supply chains. The paper makes three contributions. First, it proposes an AutoFE+TFM pipeline: it flattens a relational schema of 7 tables with Deep Feature Synthesis, and then applies in-context learning (TabICLv2, Mitra) without any fine-tuning. Second, it compares this "flatten-then-classify" approach with Griffin, a relational foundation model that keeps the structure. Third, it introduces a cold-start inference setup, where all features derived from history are masked to NaN. In this way, newly onboarded entities can still be scored using only static metadata. The experiments use about 3,700 entities and three data splits (temporal, stratified, and disjoint-supplier). The foundation models match or beat an XGBoost-on-DFS baseline in AUC and P@90R, and the gap is largest under temporal shift. The cold-start mode also keeps an AUC of .729 using metadata alone.

## Strengths
- **Well-motivated, realistic application.** The setting stresses three axes (multi-relational temporal structure, 62–81% sparsity, cold-start) that standard tabular benchmarks omit, and the workshop audience would find the case study genuinely informative.
- **Sensible evaluation design.** Three complementary splits, with the disjoint-supplier split serving as a clean cold-start test, show care in experimental setup.
- **Practical cold-start mechanism.** Scoring zero-history entities by NaN-masking history features is simple, deployable, and a useful takeaway for practitioners.
- **Several honest qualifications**, e.g., acknowledging that resolution-table gains are "modest" and that XGBoost retains higher F1 under temporal shift due to better calibration.

## Weaknesses
- **No variance or significance testing.** Headline improvements (+.003 P@90R, +.005 F1 from resolution tables) are within plausible run-to-run noise, yet no error bars, seeds, or significance tests are reported anywhere. As stated, these gains are not convincingly distinguishable from zero.
- **The central "match production systems" claim is not measured.** The experimental baseline is XGBoost on DFS features, not the hand-crafted production system. The production pipeline's actual predictive metrics are never reported head-to-head (Appendix C.1 compares only feature importances), so the headline claim is unsubstantiated.
- **Cold-start result conflates metrics.** Comparing AUC .729 to a ".614 sector prevalence prior" is apples-to-oranges, a prevalence prior has AUC 0.5. Combined with a 57% test prevalence, the cold-start P@90R lift is thinner than the framing suggests.
- **"Normalized P@90R (ref. prevalence 50%)" is never defined,** despite the cold-start argument depending on it; notation also alternates between P@90R and NP@90R.
- **TabPFNv2.5 is claimed as part of the pipeline but appears in no results table.**
- **Internal numerical inconsistencies.** §5.3 reports cold-start full-feature results as .807/.665/.750 while Table 2 lists .797/.673/.759; selected-feature counts differ (45 vs. 32) with mismatched DFS depth/primitive settings between §4.2 and Appendix C.2; and C.2 cross-references a non-existent "Section 3.1."
- **Mechanistic explanations are asserted, not tested** (e.g., that attention-based null discounting drives the FM advantage), with no ablation isolating the effect.
- **"Complementary paradigms" overstates the evidence;** the results show the two approaches perform comparably, not that they capture complementary signal.

## Suggestions
- Report mean ± std over multiple seeds (and ideally a paired significance test) for all main-table results; this is the single most important fix for the claims-vs-evidence gap.
- Add the actual hand-crafted production pipeline as a row in Table 2 on the same splits, or rephrase the comparison as "FM vs. XGBoost-on-DFS" and soften the "match production" claim accordingly.
- For cold-start, compare like-for-like: report the AUC of an appropriate baseline (e.g., metadata-only logistic/XGBoost) rather than a prevalence number, and state the test-set prevalence next to P@90R.
- Define "normalized P@90R" precisely (formula and reference prevalence) and use one consistent symbol.
- Either include TabPFNv2.5 results or remove it from the contributions/methods.
- Reconcile the text/table discrepancies and feature-count/DFS-setting mismatches; fix the §3.1 cross-reference.
- Reframe asserted mechanisms ("we hypothesize…") or add a targeted ablation; downgrade "complementary" to "comparable" unless an ensemble or win/lose analysis is added.
- Move Figure 3 to the main text: The pipeline diagram in Appendix D is highly informative. Moving it to the main text (e.g., Section 4) would greatly help readers understand the AutoFE+TFM and RFM data flows without needing to flip to the appendix.

---

### Official Review · Reviewer_y5oR · 2026-05-22
**Useful Applied Study, but Dataset Distinctiveness Needs Clearer Justification**

**Rating:** 5
**Confidence:** 3

**Review:**

## Summary

This paper studies foundation-model-based methods for sparse, multi-relational risk prediction in global supply chains. The paper compares feature-synthesis plus tabular foundation models with relational foundation models, and evaluates them under temporal and cold-start settings.

## Strengths

The comparison between feature-synthesis-based tabular models and relational foundation models is useful. It gives readers a practical view of two different ways to handle multi-table data: flattening relational information into engineered features versus modeling the relational structure more directly.

The cold-start analysis is also valuable. In many real enterprise settings, the model needs to score entities with limited historical observations, so evaluating metadata-only or low-history prediction is relevant.

## Areas for Improvement

My main concern is how clearly the paper distinguishes this dataset from existing relational database benchmarks such as RelBench and 4DBInfer. The paper motivates the global supply-chain task through multi-relational temporal structure, sparsity, and cold-start inference. However, these properties are already at least partially covered by existing RDB benchmarks, which include temporal splits, multi-table schemas, inductive settings, and graph/tabular modeling pipelines.

Therefore, the paper should more explicitly explain what is distinct about this dataset. For example, is the sparsity pattern more extreme? Are the compliance-risk objectives or operational metrics different from existing benchmark tasks? Without this comparison, it is difficult to judge whether the dataset provides a genuinely new evaluation setting or mainly serves as a domain-specific application case study.

## Detailed Comments

The authors should compare the proposed global supply-chain dataset more directly with RelBench and 4DBInfer. In particular, they should discuss whether the dataset differs in schema structure, sparsity level, temporal dynamics, cold-start ratio, task objective, or evaluation protocol.

It would be useful to report quantitative dataset statistics that support the claimed distinctiveness of the setting, such as table coverage, missingness rates, number of records per supplier, percentage of suppliers with no prior assessments, and train/test entity overlap.

It would also be helpful to discuss how the dataset or task format could be made useful for future readers who want to test methods on similar sparse multi-relational risk prediction problems.

## Justification of Score

I would rate this paper as a 5. The paper studies an important and realistic structured-data application, and the comparison between tabular foundation models and relational foundation models is useful for the workshop. However, I am not fully convinced that the dataset’s distinctiveness relative to existing RDB benchmarks is sufficiently established. Stronger positioning against RelBench and 4DBInfer, together with more quantitative dataset analysis, would make the contribution more convincing.